# POSITIVE-UNLABELED DIFFUSION MODELS FOR PREVENTING SENSITIVE DATA GENERATION

**Hiroshi Takahashi**[*]
NTT Corporation

**Tomoharu Iwata**
NTT Corporation

**Atsutoshi Kumagai**
NTT Corporation

**Yuuki Yamanaka**
NTT Corporation

**Tomoya Yamashita**
NTT Corporation

## ABSTRACT

Diffusion models are powerful generative models but often generate sensitive data that are unwanted by users, mainly because the unlabeled training data frequently contain such sensitive data. Since labeling all sensitive data in the large-scale unlabeled training data is impractical, we address this problem by using a small amount of labeled sensitive data. In this paper, we propose positive-unlabeled diffusion models, which prevent the generation of sensitive data using unlabeled and sensitive data. Our approach can approximate the evidence lower bound (ELBO) for normal (negative) data using only unlabeled and sensitive (positive) data. Therefore, even without labeled normal data, we can maximize the ELBO for normal data and minimize it for labeled sensitive data, ensuring the generation of only normal data. Through experiments across various datasets and settings, we demonstrated that our approach can prevent the generation of sensitive images without compromising image quality.

## 1 INTRODUCTION

Diffusion models (Ho et al., 2020; Sohl-Dickstein et al., 2015; Song & Ermon, 2019; Song et al., 2020b) are powerful generative models that have become the de facto standard, and are applied to various fields such as images (Dhariwal & Nichol, 2021; Rombach et al., 2022), audio (Chen et al., 2020; Kong et al., 2020; Popov et al., 2021), and text (Austin et al., 2021; Li et al., 2022; Gong et al., 2022). The training of diffusion models can be regarded as the maximization of the evidence lower bound (ELBO), which is the tractable lower bound of the log-likelihood, on the training data (Ho et al., 2020). Users collect these training data from sources like the internet to generate the contents they want, and then perform either training from scratch or fine-tuning.

Unfortunately, diffusion models have the potential to generate inappropriate, discriminatory, or harmful contents that are unwanted by users (Brack et al., 2022). For example, they might generate sexual images of real individuals (Mirsky & Lee, 2021; Verdoliva, 2020). We refer to such contents as sensitive data. The primary cause of this problem is that such sensitive data are included in the training data. To handle this problem, existing approaches have applied data forgetting (Gandikota et al., 2023; Zhang et al., 2024) and continual learning (Heng & Soh, 2024) to pre-trained diffusion models. For example, to forget sensitive data, Heng & Soh (2024) first prepare normal and sensitive data, then attempt to maximize the ELBO for the normal data while minimizing it for the sensitive data. This approach requires a lot of normal data that do not include any sensitive data. However, it is difficult to prepare such normal data. When using training data collected from the internet, it is difficult to manually remove all sensitive data. In addition, when using pre-trained models to generate training data, unintended sensitive data may be produced. That is, what we can actually use is not clean normal data, but unlabeled data that contain both normal and sensitive data. On the other hand, it is easy to prepare a small amount of sensitive data that users may not want. Hence, we need to train diffusion models in positive-unlabeled (PU) setting (Du Plessis et al., 2014; 2015; Kiryo et al., 2017), where we have access only to unlabeled and sensitive (positive) data, but not to normal (negative) data.

---

[*]Corresponding author: `hiroshibm.takahashi@ntt.com`

(a) Unlabeled training data. (b) Sensitive training data. (c) Unsupervised samples. (d) Proposed samples.

Figure 1: MNIST examples by the proposed method, where even numbers are considered normal data and odd numbers are considered sensitive data. (a) The unlabeled training data contain 10% sensitive data (odd numbers). (b) The sensitive training data contain only odd numbers. (c) When the diffusion model is trained in the standard way using the unlabeled training data, the generated samples include sensitive data (odd numbers). (d) When the proposed method is applied to the diffusion model, it generates only normal data (even numbers).

In this paper, we propose positive-unlabeled diffusion models, which prevent diffusion models from generating sensitive data for this PU setting. We model an unlabeled data distribution with a mixture of normal and sensitive data distributions. Accordingly, the normal data distribution can be rewritten as a mixture of unlabeled and sensitive data distributions. With this trick, we approximate the ELBO for normal data only using unlabeled and sensitive data. Therefore, even without labeled normal data, we can maximize the ELBO for normal data and minimize it for labeled sensitive data. Note that our approach requires only a small amount of labeled sensitive data because such data are less diverse than normal data.

Figure 1 shows MNIST examples by unsupervised and proposed methods. In these examples, even numbers are considered normal data, while odd numbers are considered sensitive data. Our approach can maximize the ELBO for normal data (even numbers) and minimize it for sensitive data (odd numbers), using only unlabeled and sensitive data. As a result, the diffusion model trained with our approach generates only normal data (even numbers).

Our approach can be applied to training from scratch as well as to fine-tuning by using the parameters of a pre-trained model as initial values. Through experiments across various datasets and settings, we demonstrated that our approach effectively prevents the generation of sensitive images without compromising image quality.

Our approach can also be easily extended to positive-negative-unlabeled setting, where we can use a small amount of normal data. Our approach is well-suited for privacy-preserving applications, ensuring that sensitive samples are not inadvertently generated while maintaining sample quality. The code is available at ⌂ https://github.com/takahashihiroshi/pudm.

Our contributions can be summarized as follows:

- We propose positive-unlabeled diffusion models, designed to prevent diffusion models from generating sensitive data when only unlabeled and sensitive data are available.
- Our approach supports both training from scratch or fine-tuning pre-trained models.
- Experiments on various datasets show that our approach effectively prevents the generation of sensitive images without compromising image quality.

## 2 PRELIMINARIES

### 2.1 PROBLEM SETUP

First, we describe our problem setup. Given unlabeled dataset $\mathcal{U} = \{\mathbf{x}^{(1)}, \ldots, \mathbf{x}^{(N)}\}$ and labeled sensitive dataset $\mathcal{S} = \{\tilde{\mathbf{x}}^{(1)}, \ldots, \tilde{\mathbf{x}}^{(M)}\}$ for training, $\mathcal{U}$ contains both normal and sensitive data points. Our goal is to train diffusion models so that they generate only normal data using $\mathcal{U}$ and $\mathcal{S}$.

## 2.2 Diffusion Models

Next, we review diffusion models (Ho et al., 2020; Sohl-Dickstein et al., 2015; Song & Ermon, 2019; Song et al., 2020b). They consist of two processes: diffusion and denoising.

The diffusion process gradually adds Gaussian noise to the original data point $\mathbf{x}_0$ over $T$ steps, eventually converting it into standard Gaussian noise $\mathbf{x}_T$. The denoising process gradually removes the noise from $\mathbf{x}_T$, converting it back to the original data point $\mathbf{x}_0$. The noisy data points $\mathbf{x}_1, \ldots, \mathbf{x}_T$ can be viewed as latent variables of the same dimension as the original data point $\mathbf{x}_0$.

They model the probability of a data point $\mathbf{x}_0$ using latent variables $\mathbf{x}_1, \ldots, \mathbf{x}_T$ as follows:

$$p_\theta(\mathbf{x}_0) = \int p_\theta(\mathbf{x}_{0:T}) d\mathbf{x}_{1:T}, \tag{1}$$

where $p_\theta$ represents the denoising process:

$$p_\theta(\mathbf{x}_{0:T}) = p(\mathbf{x}_T) \prod_{t=1}^{T} p_\theta(\mathbf{x}_{t-1}|\mathbf{x}_t). \tag{2}$$

These distributions are modeled by Gaussian distributions:

$$p_\theta(\mathbf{x}_{t-1}|\mathbf{x}_t) = \mathcal{N}(\mathbf{x}_{t-1}; \mu_\theta(\mathbf{x}_t, t), \Sigma_\theta(\mathbf{x}_t, t)), \quad p(\mathbf{x}_T) = \mathcal{N}(\mathbf{x}_T; \mathbf{0}, \mathbf{I}). \tag{3}$$

Here, $\mu_\theta(\mathbf{x}_t, t)$ and $\Sigma_\theta(\mathbf{x}_t, t)$ are neural networks with parameter $\theta$, which estimate the mean and covariance matrix of the Gaussian distribution, respectively. $\mathcal{N}(\mathbf{x}_T; \mathbf{0}, \mathbf{I})$ is the standard Gaussian, where $\mathbf{0}$ denotes the zero vector and $\mathbf{I}$ denotes the identity matrix.

On the other hand, the diffusion process is modeled by Gaussian distributions as follows:

$$q(\mathbf{x}_{1:T}|\mathbf{x}_0) = \prod_{t=1}^{T} q(\mathbf{x}_t|\mathbf{x}_{t-1}), \quad q(\mathbf{x}_t|\mathbf{x}_{t-1}) = \mathcal{N}(\mathbf{x}_t; \sqrt{\alpha_t}\mathbf{x}_{t-1}, (1-\alpha_t)\mathbf{I}), \tag{4}$$

where $\alpha_t \in (0, 1)$ is a monotonically decreasing hyperparameter. A notable property of the diffusion process is that $\mathbf{x}_t$ at an arbitrary step $t$ can be sampled in closed form:

$$q(\mathbf{x}_t|\mathbf{x}_0) = \mathcal{N}(\mathbf{x}_t; \sqrt{\bar{\alpha}_t}\mathbf{x}_0, (1-\bar{\alpha}_t)\mathbf{I}), \quad \bar{\alpha}_t = \prod_{t'=1}^{t} \alpha_{t'}. \tag{5}$$

By applying the reparameterization trick (Kingma, 2013), $\mathbf{x}_t$ can be rewritten as follows:

$$\mathbf{x}_t = \sqrt{\bar{\alpha}_t}\mathbf{x}_0 + \sqrt{1-\bar{\alpha}_t}\epsilon, \tag{6}$$

where $\epsilon$ is a sample drawn from a standard Gaussian $\mathcal{N}(\epsilon; \mathbf{0}, \mathbf{I})$.

With these distributions, the ELBO for each data point $\mathbf{x}_0$, $\mathcal{L}_{\mathrm{DM}}(\mathbf{x}_0; \theta)$, can be derived as follows:

$$\log p_\theta(\mathbf{x}_0) = \log \mathbb{E}_{q(\mathbf{x}_{1:T}|\mathbf{x}_0)}\left[\frac{p_\theta(\mathbf{x}_{0:T})}{q(\mathbf{x}_{1:T}|\mathbf{x}_0)}\right] \geq \mathbb{E}_{q(\mathbf{x}_{1:T}|\mathbf{x}_0)}\left[\log \frac{p_\theta(\mathbf{x}_{0:T})}{q(\mathbf{x}_{1:T}|\mathbf{x}_0)}\right] \equiv \mathcal{L}_{\mathrm{DM}}(\mathbf{x}_0; \theta), \tag{7}$$

where $\mathbb{E}[\cdot]$ is the expectation. This ELBO can be simplified to the following objective function:

$$-\mathcal{L}_{\mathrm{DM}}(\mathbf{x}_0; \theta) \propto \mathbb{E}_{u(t), p(\epsilon)}\left[\left\|\epsilon - \epsilon_\theta\left(\sqrt{\bar{\alpha}_t}\mathbf{x}_0 + \sqrt{1-\bar{\alpha}_t}\epsilon, t\right)\right\|^2\right] \equiv \ell(\mathbf{x}_0; \theta), \tag{8}$$

where $u(t)$ is a uniform distribution over 1 to $T$, $p(\epsilon)$ is a standard Gaussian $\mathcal{N}(\epsilon; \mathbf{0}, \mathbf{I})$, and $\epsilon_\theta$ is a neural network that estimates the noise $\epsilon$ from $\mathbf{x}_t = \sqrt{\bar{\alpha}_t}\mathbf{x}_0 + \sqrt{1-\bar{\alpha}_t}\epsilon$ at step $t$. That is, we can train diffusion models by minimizing the squared error between the noise $\epsilon$ and the estimated noise $\epsilon_\theta(\mathbf{x}_t, t)$. Note that the ELBO is maximized by minimizing $\ell(\mathbf{x}_0; \theta)$, and conversely, the ELBO is minimized by maximizing it.

In standard diffusion model training, all unlabeled data points in $\mathcal{U}$ are assumed to be normal, and the following objective function is minimized:

$$\frac{1}{N}\sum_{n=1}^{N} \ell(\mathbf{x}^{(n)}; \theta). \tag{9}$$

However, since the unlabeled data often contain sensitive data, the model trained with the above objective may generate sensitive data.

## 3 PROPOSED METHOD

In this section, we propose positive-unlabeled diffusion models, preventing diffusion models from learning sensitive data using $\mathcal{U}$ and $\mathcal{S}$. Our approach is based on PU learning that aims to train a binary classifier to distinguish between positive and negative data using positive and unlabeled data (Du Plessis et al., 2014; 2015; Kiryo et al., 2017). Hereafter, we will also refer to normal data points as negative (-) samples and sensitive data points as positive (+) samples.

### 3.1 SUPERVISED DIFFUSION MODELS

First, assuming that both normal and sensitive data are given, we discuss a supervised approach for training a diffusion model. For example, as described in (Heng & Soh, 2024), we can simply minimize $\ell(\mathbf{x}; \theta)$ for the normal data and maximize it for the sensitive data. However, since $\ell(\mathbf{x}; \theta)$ is bounded below but unbounded above, maximizing $\ell(\mathbf{x}; \theta)$ will cause it to diverge to infinity, resulting in meaningless parameters. Although $1/\ell(\mathbf{x}; \theta)$ can be minimized as an alternative, this requires an additional hyperparameter that balances $\ell(\mathbf{x}; \theta)$ and $1/\ell(\mathbf{x}; \theta)$, as described in (Ruff et al., 2019).

As a reasonable supervised approach, following (Yamanaka et al., 2019), we introduce a binary classification framework into diffusion model training. Let $y = 0$ represent normal data and $y = 1$ represent sensitive data. We model the conditional probability $p_\theta(y|\mathbf{x})$ using $\ell(\mathbf{x}; \theta)$ as follows:

$$p_\theta(y|\mathbf{x}) = \begin{cases} \exp(-\ell(\mathbf{x}; \theta)) & (y = 0) \\ 1 - \exp(-\ell(\mathbf{x}; \theta)) & (y = 1). \end{cases} \tag{10}$$

A small $\ell(\mathbf{x}; \theta)$ leads to a higher $p_\theta(y = 0|\mathbf{x})$, while a large $\ell(\mathbf{x}; \theta)$ results in a higher $p_\theta(y = 1|\mathbf{x})$. With this conditional probability, we introduce the binary cross entropy as the loss function for each data point as follows:

$$\ell_{\mathrm{BCE}}(\mathbf{x}, y; \theta) = -\log p_\theta(y|\mathbf{x}) = (1 - y)\ell(\mathbf{x}; \theta) - y\log(1 - \exp(-\ell(\mathbf{x}; \theta))). \tag{11}$$

This loss function minimizes $\ell(\mathbf{x}; \theta)$ for $y = 0$, and maximizes it for $y = 1$ thorough minimizing $-\log(1 - \exp(-\ell(\mathbf{x}; \theta)))$. Since these two terms are bounded below, there is no risk of divergence. In addition, as described in (Yamanaka et al., 2019), they are well-balanced, eliminating the need for additional hyperparameters. Further theoretical justification is provided in Appendix.

In a supervised approach, we assume that all unlabeled data points in $\mathcal{U}$ are normal, and minimize the following objective function:

$$\frac{1}{N} \sum_{n=1}^{N} \ell_{\mathrm{BCE}}(\mathbf{x}^{(n)}, 0; \theta) + \frac{1}{M} \sum_{m=1}^{M} \ell_{\mathrm{BCE}}(\widetilde{\mathbf{x}}^{(m)}, 1; \theta). \tag{12}$$

This supervised approach is expected to perform better than the unsupervised approach (Eq. (9)) because it can handle sensitive data $\mathcal{S}$. However, the presence of sensitive data in the unlabeled data $\mathcal{U}$ weakens the effect of maximizing $\ell(\mathbf{x}; \theta)$ for sensitive data.

### 3.2 POSITIVE-UNLABELED DIFFUSION MODELS

To handle the unlabeled data $\mathcal{U}$ that contain sensitive data, we introduce a PU learning framework into the supervised diffusion model. Let $p_{\mathcal{U}}(\mathbf{x})$ represent the unlabeled data distribution, $p_{\mathcal{S}}(\mathbf{x})$ represent the sensitive data distribution, and $p_{\mathcal{N}}(\mathbf{x})$ represent the normal data distribution. The datasets $\mathcal{U}$ and $\mathcal{S}$ are assumed to be drawn from $p_{\mathcal{U}}(\mathbf{x})$ and $p_{\mathcal{S}}(\mathbf{x})$, respectively. We model $p_{\mathcal{U}}(\mathbf{x})$ with a mixture of $p_{\mathcal{S}}(\mathbf{x})$ and $p_{\mathcal{N}}(\mathbf{x})$:

$$p_{\mathcal{U}}(\mathbf{x}) = \beta p_{\mathcal{S}}(\mathbf{x}) + (1 - \beta)p_{\mathcal{N}}(\mathbf{x}), \tag{13}$$

where $\beta \in [0, 1]$ represents the ratio of the sensitive data in the unlabeled data $\mathcal{U}$. Accordingly, $p_{\mathcal{N}}(\mathbf{x})$ can be rewritten as follows:

$$(1 - \beta)p_{\mathcal{N}}(\mathbf{x}) = p_{\mathcal{U}}(\mathbf{x}) - \beta p_{\mathcal{S}}(\mathbf{x}). \tag{14}$$

Note that we can estimate the hyperparameter $\beta$ by PU learning approaches (Menon et al., 2015; Ramaswamy et al., 2016; Jain et al., 2016; Christoffel et al., 2016; Nakajima & Sugiyama, 2023).

---

**Algorithm 1** Positive-Unlabeled Diffusion Model with Stochastic Gradient Descent

---

**Require:** unlabeled and sensitive datasets $(\mathcal{U}, \mathcal{S})$, hyperparameter $\beta \in [0, 1]$
**Ensure:** parameter of diffusion model $\theta$
 1: **while** not converged **do**
 2:     Sample mini-batch $\mathcal{B}$ from datasets $(\mathcal{U}, \mathcal{S})$
 3:     Compute $\mathcal{L}_{\mathcal{S}}^{+}(\theta)$, $\mathcal{L}_{\mathcal{U}}^{-}(\theta)$, and $\mathcal{L}_{\mathcal{S}}^{-}(\theta)$ in Eq. (18) with $\mathcal{B}$
 4:     **if** $\mathcal{L}_{\mathcal{U}}^{-}(\theta) - \beta\mathcal{L}_{\mathcal{S}}^{-}(\theta) \geq 0$ **then**
 5:         Compute gradient $\nabla_{\theta}(\beta\mathcal{L}_{\mathcal{S}}^{+}(\theta) + \mathcal{L}_{\mathcal{U}}^{-}(\theta) - \beta\mathcal{L}_{\mathcal{S}}^{-}(\theta))$
 6:     **else**
 7:         Compute gradient $\nabla_{\theta} - (\mathcal{L}_{\mathcal{U}}^{-}(\theta) - \beta\mathcal{L}_{\mathcal{S}}^{-}(\theta))$
 8:     **end if**
 9:     Update $\theta$ with the above gradient
10: **end while**

---

If we have access to $p_{\mathcal{N}}(\mathbf{x})$, we can train the diffusion model by minimizing the following supervised objective function:

$$\mathcal{L}_{\mathrm{PN}}(\theta) = \beta\mathbb{E}_{p_{\mathcal{S}}}[\ell_{\mathrm{BCE}}(\mathbf{x}, 1; \theta)] + (1 - \beta)\mathbb{E}_{p_{\mathcal{N}}}[\ell_{\mathrm{BCE}}(\mathbf{x}, 0; \theta)]. \tag{15}$$

Since we do not have access to $p_{\mathcal{N}}(\mathbf{x})$ in practice, we compute the second term in Eq. (15) using Eq. (14) as follows:

$$(1 - \beta)\mathbb{E}_{p_{\mathcal{N}}}[\ell_{\mathrm{BCE}}(\mathbf{x}, 0; \theta)] = \mathbb{E}_{p_{\mathcal{U}}}[\ell_{\mathrm{BCE}}(\mathbf{x}, 0; \theta)] - \beta\mathbb{E}_{p_{\mathcal{S}}}[\ell_{\mathrm{BCE}}(\mathbf{x}, 0; \theta)]. \tag{16}$$

Hence, only using $p_{\mathcal{U}}(\mathbf{x})$ and $p_{\mathcal{S}}(\mathbf{x})$, we can compute $\mathcal{L}_{\mathrm{PN}}(\theta)$ as follows:

$$\mathcal{L}_{\mathrm{PN}}(\theta) = \beta\mathbb{E}_{p_{\mathcal{S}}}[\ell_{\mathrm{BCE}}(\mathbf{x}, 1; \theta)] + \mathbb{E}_{p_{\mathcal{U}}}[\ell_{\mathrm{BCE}}(\mathbf{x}, 0; \theta)] - \beta\mathbb{E}_{p_{\mathcal{S}}}[\ell_{\mathrm{BCE}}(\mathbf{x}, 0; \theta)]. \tag{17}$$

We approximate $\mathcal{L}_{\mathrm{PN}}(\theta)$ using the datasets $\mathcal{U}$ and $\mathcal{S}$ as follows:

$$\mathcal{L}_{\mathrm{PN}}(\theta) \simeq \beta\underbrace{\frac{1}{M}\sum_{m=1}^{M}\ell_{\mathrm{BCE}}(\tilde{\mathbf{x}}^{(m)}, 1; \theta)}_{\mathcal{L}_{\mathcal{S}}^{+}(\theta)} + \underbrace{\frac{1}{N}\sum_{n=1}^{N}\ell_{\mathrm{BCE}}(\mathbf{x}^{(n)}, 0; \theta)}_{\mathcal{L}_{\mathcal{U}}^{-}(\theta)} - \beta\underbrace{\frac{1}{M}\sum_{m=1}^{M}\ell_{\mathrm{BCE}}(\tilde{\mathbf{x}}^{(m)}, 0; \theta)}_{\mathcal{L}_{\mathcal{S}}^{-}(\theta)}. \tag{18}$$

In Eq. (18), $\mathcal{L}_{\mathcal{U}}^{-}(\theta) - \beta\mathcal{L}_{\mathcal{S}}^{-}(\theta)$ is the approximation of $(1-\beta)\mathbb{E}_{p_{\mathcal{N}}}[\ell_{\mathrm{BCE}}(\mathbf{x}, 0; \theta)] \geq 0$. Unfortunately, $\mathcal{L}_{\mathcal{U}}^{-}(\theta) - \beta\mathcal{L}_{\mathcal{S}}^{-}(\theta) \geq 0$ does not always hold, which can lead to over-fitting (Kiryo et al., 2017). To avoid this, our training objective function ensures $\mathcal{L}_{\mathcal{U}}^{-}(\theta) - \beta\mathcal{L}_{\mathcal{S}}^{-}(\theta) \geq 0$ according to Kiryo et al. (2017) as follows:

$$\mathcal{L}_{\mathrm{PU}}(\theta) = \beta\mathcal{L}_{\mathcal{S}}^{+}(\theta) + \max\left\{0, \mathcal{L}_{\mathcal{U}}^{-}(\theta) - \beta\mathcal{L}_{\mathcal{S}}^{-}(\theta)\right\}. \tag{19}$$

We optimize this objective using the stochastic optimization method such as AdamW (Loshchilov, 2017). In practice, when $\mathcal{L}_{\mathcal{U}}^{-}(\theta) - \beta\mathcal{L}_{\mathcal{S}}^{-}(\theta) < 0$, we maximize it until $\mathcal{L}_{\mathcal{U}}^{-}(\theta) - \beta\mathcal{L}_{\mathcal{S}}^{-}(\theta) \geq 0$, according to Kiryo et al. (2017). Algorithm 1 provides the pseudocode for our approach.

Alternatively, instead of the max value $\max\left\{0, \mathcal{L}_{\mathcal{U}}^{-}(\theta) - \beta\mathcal{L}_{\mathcal{S}}^{-}(\theta)\right\}$ in Eq. (19), we can use the absolute value $\left|\mathcal{L}_{\mathcal{U}}^{-}(\theta) - \beta\mathcal{L}_{\mathcal{S}}^{-}(\theta)\right|$, according to (Hammoudeh & Lowd, 2020). This is effective for more complex models such as Stable Diffusion (Rombach et al., 2022), as shown in Appendix.

## 3.3 EXTENSION TO CONDITIONAL DIFFUSION MODELS

While the above discussion has focused on unconditional diffusion models, our approach can be easily extended to conditional diffusion models. Given a condition $\mathbf{c}$ such as text, it is sufficient to replace the ELBO in Eq. (8) with the following ELBO:

$$\ell(\mathbf{x}_0, \mathbf{c}; \theta) \equiv \mathbb{E}_{u(t), p(\epsilon)}\left[\left\|\epsilon - \epsilon_{\theta}\left(\sqrt{\bar{\alpha}_t}\mathbf{x}_0 + \sqrt{1 - \bar{\alpha}_t}\epsilon, \mathbf{c}, t\right)\right\|^2\right], \tag{20}$$

where $\epsilon_{\theta}$ is a neural network that estimates the noise $\epsilon$ from $\mathbf{x}_t = \sqrt{\bar{\alpha}_t}\mathbf{x}_0 + \sqrt{1 - \bar{\alpha}_t}\epsilon$, condition $\mathbf{c}$ and step $t$. We can efficiently draw samples from this conditional model by using classifier-free guidance (Ho & Salimans, 2022).

# 4 RELATED WORK

## 4.1 ERASING SENSITIVE CONCEPTS IN DIFFUSION MODELS

Several approaches have tried to prevent conditional diffusion models from generating sensitive images (Brack et al., 2022; Gandikota et al., 2023; Zhang et al., 2024; Heng & Soh, 2024). For example, Heng & Soh (2024) aim to maximize the ELBO for normal data and minimize it for sensitive data within the continual learning framework (Kirkpatrick et al., 2017; Shin et al., 2017). However, this has been experimentally shown to compromise image quality (Heng & Soh, 2024). One reason is that the ELBO is unbounded below and can easily diverge to negative infinity [1], as discussed in Section 3.1. To address this issue, for the sensitive data point $\tilde{\mathbf{x}}$ and its condition $\tilde{\mathbf{c}}$, Heng & Soh (2024) replace $\tilde{\mathbf{x}}$ with completely different data $\tilde{\mathbf{z}}$, such as noise following a uniform distribution, and maximize the ELBO for $\tilde{\mathbf{z}}$ and $\tilde{\mathbf{c}}$. As a result, When the condition $\tilde{\mathbf{c}}$ is input, the fine-tuned model generates $\tilde{\mathbf{z}}$ instead of the sensitive data $\tilde{\mathbf{x}}$. Although this approach is effective, it can only be applied to conditional models. Moreover, it cannot work well in PU setting because it assumes all unlabeled data are normal. The same limitation applies to other approaches (Brack et al., 2022; Gandikota et al., 2023; Zhang et al., 2024) as well.

Compared to existing approaches, our objective function reasonably minimizes the ELBO for the sensitive data while preventing divergence to negative infinity, as described in Section 3.1. This enables us to avoid compromising image quality. Furthermore, our approach can be applied to both unconditional and conditional models, as well as PU setting we desire.

## 4.2 POSITIVE-UNLABELED LEARNING AND ANOMALY DETECTION

Our approach is closely related to PU learning, which trains a binary classifier to distinguish between positive and negative data using positive and unlabeled data. Our approach is based on the unbiased PU learning (Du Plessis et al., 2014; 2015; Kiryo et al., 2017), and has the ideal property that Eq. (18) converges to Eq. (15) as the dataset sizes $N, M \to \infty$.

However, since PU learning is designed for binary classification, it cannot be directly extended to diffusion models. To address this limitation, our approach first incorporates a binary classification framework into diffusion model training according to Yamanaka et al. (2019), and then applies PU learning to this framework. A similar approach has been employed in the context of semi-supervised anomaly detection (Takahashi et al., 2024), which aims to train the anomaly detector using the unlabeled and anomaly data. Although this approach has a similar problem setting to ours, it lacks generative capabilities because it is based on autoencoders (Hinton & Salakhutdinov, 2006). To the best of our knowledge, our approach is the first to train diffusion models in PU setting.

# 5 EXPERIMENTS

## 5.1 DATA

We used the following image datasets: MNIST (LeCun et al., 1998), CIFAR10 (Krizhevsky et al., 2009), STL10 (Coates et al., 2011), and CelebA (Liu et al., 2015). MNIST is handwritten digits, CIFAR10 and STL10 contain images of animals and vehicles, and CelebA is celebrity faces. We converted MNIST to $32 \times 32$ three-channel images, and CelebA to $256 \times 256$ images.

Each dataset was divided into two categories: MNIST into even and odd numbers, CIFAR10 and STL10 into animals and vehicles, and CelebA into male and female. We treated one category as normal, and the other as sensitive. From these datasets, we prepared training and test data as follows. The training data consist of unlabeled data $\mathcal{U}$ and labeled sensitive data $\mathcal{S}$, where $\mathcal{U}$ include both normal and sensitive data. Meanwhile, the test data contain only normal data. For example, in MNIST, if we treat even numbers as normal, the unlabeled training data $\mathcal{U}$ include both even and odd numbers, the sensitive training data $\mathcal{S}$ include only odd numbers, and the test data include only even numbers. The number of data points in each dataset is shown in Table 1.

---

[1]Note that the ELBO is unbounded below, and the negative ELBO $\ell(\mathbf{x}_0; \theta)$ in Eq. (8) is unbounded above.

Table 1: Number of data points in each dataset. The parentheses next to the dataset names indicate the normal class.

|  | Image size | $\mathcal{U}$ (normal) | $\mathcal{U}$ (sensitive) | $\mathcal{S}$ | Test |
|---|---|---|---|---|---|
| MNIST (even) | $32 \times 32$ | 25,000 | 2,500 | 2,500 | 4,926 |
| MNIST (odd) | $32 \times 32$ | 25,000 | 2,500 | 2,500 | 5,074 |
| CIFAR10 (vehicles) | $32 \times 32$ | 20,000 | 2,000 | 2,000 | 4,000 |
| CIFAR10 (animals) | $32 \times 32$ | 20,000 | 2,000 | 2,000 | 6,000 |
| STL10 (vehicles) | $96 \times 96$ | 2,000 | 200 | 200 | 3,200 |
| STL10 (animals) | $96 \times 96$ | 2,000 | 200 | 200 | 4,800 |
| CelebA (male) | $256 \times 256$ | 2,000 | 200 | 200 | 7,715 |
| CelebA (female) | $256 \times 256$ | 2,000 | 200 | 200 | 12,247 |

## 5.2 METRIC

We used a custom-defined metric called non-sensitive rate and the Fréchet Inception Distance (FID) score (Heusel et al., 2017) as evaluation metrics. The non-sensitive rate is the ratio of the generated samples classified as normal (non-sensitive) by a pre-trained classifier. It equals one if all generated samples belong to the normal class and zero if they all belong to the sensitive class. We use this metric to evaluate the frequency with which the diffusion model generates sensitive data. As the classifier, we used a convolutional neural network (LeCun et al., 1998) for MNIST, and used a pre-trained ResNet-34 (He et al., 2016) for the other datasets. We trained or fine-tuned these classifiers on each dataset by solving a classification task. The accuracies on the test set for the classification task are as follows: MNIST: 98.46%, CIFAR10: 95.8%, STL10: 99.5%, and CelebA: 98.45%. The FID is used to evaluate the generated samples, with a lower score indicating a better generative model. We calculated the FID between the samples generated by the diffusion model and the test data. The number of generated samples was set to be the same as the number of test data points.

## 5.3 SETUP

We compared our approach with the following methods: the unsupervised method, which minimizes the diffusion model objective $\ell(\mathbf{x}_0; \theta)$ in Eq. (8) for unlabeled data $\mathcal{U}$ as in Eq. (9), and the supervised method, which minimizes $\ell(\mathbf{x}_0; \theta)$ for $\mathcal{U}$ and maximizes it for sensitive data $\mathcal{S}$ as in Eq. (12).

We used the U-Net (Ronneberger et al., 2015) as the noise estimator $\epsilon_\theta$. We performed from-scratch training using MNIST, CIFAR10, and STL10, and fine-tuned pre-trained models using CIFAR10 and CelebA. Our implementations are based on Diffusers (von Platen et al., 2022). For the pre-trained models, we used the available models on Diffusers for CIFAR10[2] and CelebA[3]. The details of architecture are provided in Appendix.

We optimized these models using AdamW (Loshchilov, 2017) and a cosine scheduler with warmup. We set the learning rate to $10^{-4}$, and set the warmup steps to 500. The batch size was 128 for MNIST and CIFAR10, 32 for STL10, and 16 for CelebA. The number of epochs was set to 100 for from-scratch training and 20 for fine-tuning. We set the number of steps $T$ during training to $1,000$. For sampling, we used the denoising diffusion probabilistic model (DDPM) scheduler (Ho et al., 2020) in the from-scratch training, while we used the denoising diffusion implicit model (DDIM) scheduler (Song et al., 2020a) in the fine-tuning the pre-trained models, as the pre-trained models are large and time-consuming to sample. We set the sampling steps to $1,000$ in the DDPM, and to 50 in the DDIM. For the proposed method, we set $\beta = 0.1$.

We used two machines for the experiments: one with Intel Xeon Platinum 8360Y CPU, 512GB of memory, and NVIDIA A100 SXM4 GPU, and the other with Intel Xeon Gold 6148 CPU, 384GB of memory, and NVIDIA V100 SXM2 GPU. We ran all experiments five times while changing the random seeds.

---

[2]https://huggingface.co/google/ddpm-cifar10-32
[3]https://huggingface.co/google/ddpm-celebahq-256

Table 2: Comparison of non-sensitive rates for diffusion models with from-scratch training. The parentheses next to the dataset names indicate the normal class.

|  | Unsupervised | Supervised | Proposed |
|---|---|---|---|
| MNIST (even) | 0.904±0.003 | 0.908±0.003 | **0.978±0.003** |
| MNIST (odd) | 0.900±0.006 | 0.901±0.006 | **0.975±0.004** |
| CIFAR10 (vehicles) | 0.762±0.006 | 0.760±0.004 | **0.842±0.006** |
| CIFAR10 (animals) | 0.917±0.004 | 0.918±0.004 | **0.967±0.001** |
| STL10 (vehicles) | 0.824±0.011 | 0.822±0.008 | **0.944±0.010** |
| STL10 (animals) | 0.858±0.003 | 0.863±0.014 | **0.947±0.007** |

Table 3: Comparison of FID scores for diffusion models with from-scratch training. The parentheses next to the dataset names indicate the normal class.

|  | Unsupervised | Supervised | Proposed |
|---|---|---|---|
| MNIST (even) | 3.530±0.137 | **3.469±0.130** | 4.969±0.537 |
| MNIST (odd) | **3.379±0.034** | **3.366±0.058** | 5.718±0.652 |
| CIFAR10 (vehicles) | 34.400±0.622 | 34.310±0.424 | **32.463±0.970** |
| CIFAR10 (animals) | **34.448±0.504** | 34.613±0.530 | 39.586±0.573 |
| STL10 (vehicles) | **122.797±3.539** | **122.267±2.114** | **120.633±4.256** |
| STL10 (animals) | **135.999±2.203** | **135.324±1.309** | 149.775±1.522 |

Table 4: Comparison of non-sensitive rates for fine-tuned diffusion models. The parentheses next to the dataset names indicate the normal class.

|  | Pre-trained | Unsupervised | Supervised | Proposed |
|---|---|---|---|---|
| CIFAR10 (vehicles) | 0.318±0.004 | 0.741±0.006 | 0.763±0.006 | **0.858±0.009** |
| CIFAR10 (animals) | 0.681±0.003 | 0.876±0.006 | 0.892±0.005 | **0.954±0.005** |
| CelebA (male) | 0.249±0.003 | 0.866±0.023 | 0.868±0.020 | **0.973±0.010** |
| CelebA (female) | 0.751±0.004 | 0.905±0.012 | 0.907±0.007 | **0.942±0.011** |

Table 5: Comparison of FID scores for fine-tuned diffusion models. The parentheses next to the dataset names indicate the normal class.

|  | Pre-trained | Unsupervised | Supervised | Proposed |
|---|---|---|---|---|
| CIFAR10 (vehicles) | 84.066±0.403 | 26.710±0.367 | 24.458±0.387 | **18.977±0.381** |
| CIFAR10 (animals) | 35.055±0.391 | **20.313±0.338** | **19.528±0.277** | **19.415±0.915** |
| CelebA (male) | 100.256±0.261 | **43.772±2.447** | **43.465±2.509** | 46.877±5.239 |
| CelebA (female) | 43.721±0.217 | **30.520±1.333** | **30.569±1.166** | 39.131±1.961 |

## 5.4 RESULTS

### 5.4.1 COMPARISON OF FID AND NON-SENSITIVE RATE

Tables 2 and 3 show the non-sensitive rate and FID for diffusion models with from-scratch training, respectively. Similarly, Tables 4 and 5 show the non-sensitive rate and FID for fine-tuned diffusion models. The parentheses next to the dataset names indicate the normal class. The values before $\pm$ represent the mean, and the values after $\pm$ represent the standard deviation. We used bold to highlight the best results and statistically non-different results according to a pair-wise $t$-test. We used 5% as the p-value. Note that a higher non-sensitive rate is better, while a smaller FID is better.

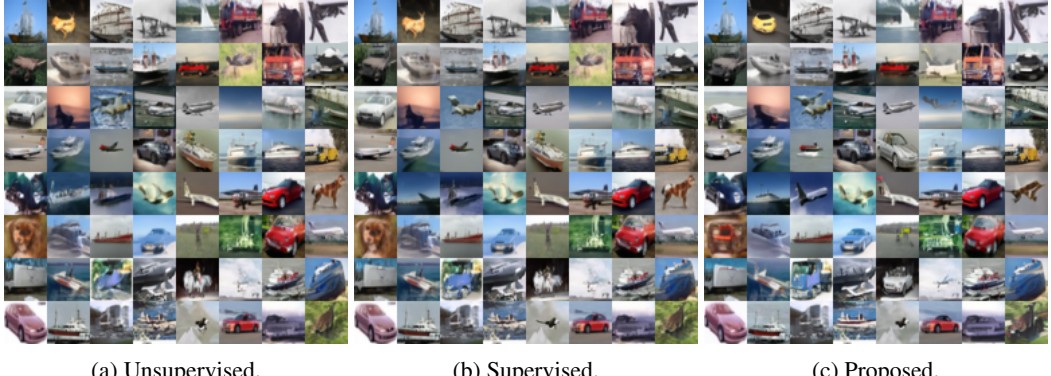

| (a) Unsupervised. | (b) Supervised. | (c) Proposed. |

Figure 2: Generated samples from fine-tuned diffusion models on CIFAR10 (vehicles).

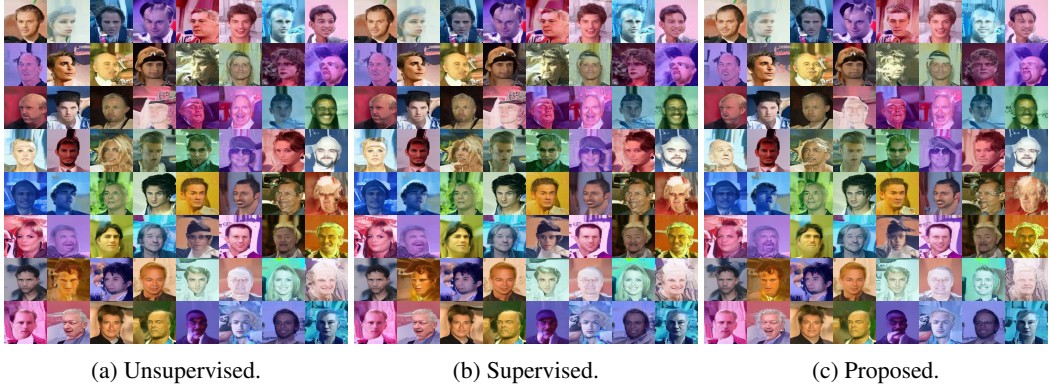

| (a) Unsupervised. | (b) Supervised. | (c) Proposed. |

Figure 3: Generated samples from fine-tuned diffusion models on CelebA (male).

First, we focus on Table 2. Since the ratio of normal data to sensitive data in the unlabeled data $\mathcal{U}$ is $10 : 1$, it is expected that the unsupervised method would generate normal data with approximately 91% probability and sensitive data with 9% probability. As shown in Table 2, the probability of generating normal data ranges from a maximum of 91.7% to a minimum of 76.2% across all datasets. In other words, assuming the classifier is perfect, at least approximately 8.3% of the generated samples are sensitive. While we would expect that supervised method improves the non-sensitive rate, unfortunately, the results remain similar to those of the unsupervised method. This indicates that supervised methods do not work well when sensitive data are included in the unlabeled data $\mathcal{U}$. On the other hand, the proposed method shows an improvement in non-sensitive rate across all datasets. This demonstrates that the proposed method can reduce the probability of generating sensitive data.

Next, we focus on Table 3. The FID is calculated between the generated samples and the test data, which contain only normal data. It is expected that if the diffusion model generates only normal data, the FID would be improved. On the other hand, learning sensitive data could improve the generation quality simply due to the increased amount of data, which might result in a better FID. As shown in Table 3, there is almost no significant difference in FID between the unsupervised and supervised methods, while the proposed method shows a comparable or slight deterioration in FID. The reason for this will be discussed later. What is important is that the deterioration in FID is extremely small. This indicates that the proposed method can prevent the generation of sensitive data without compromising the image quality.

Finally, we focus on Tables 4 and 5. Note that in the fine-tuning experiments, we also show the non-sensitive rate and FID of the pre-trained models. CIFAR10 and CelebA datasets are imbalanced; CIFAR10 has more animal images, and CelebA has more female images. As a result, the pre-trained models show a large difference in the probability of generating one class over the other. By applying

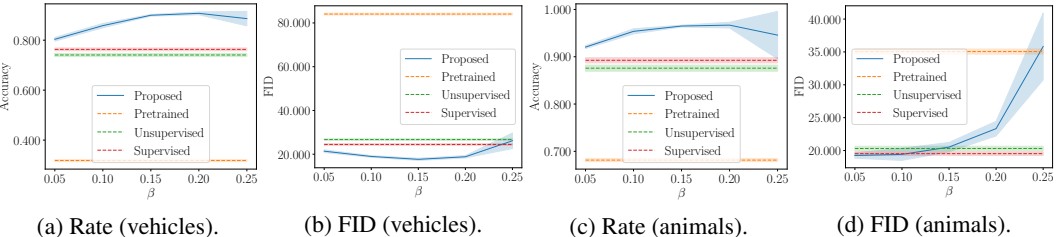

| (a) Rate (vehicles). | (b) FID (vehicles). | (c) Rate (animals). | (d) FID (animals). |

Figure 4: Non-sensitive rates and FID scores on CIFAR10 vehicles and animals with various $\beta$.

the unsupervised method, we observed an improvement in non-sensitive rate compared to the pre-trained model. As with the from-scratch training, the results of the unsupervised and supervised methods are nearly identical. On the other hand, the proposed method significantly improves non-sensitive rate. In terms of the FID, using the unsupervised, supervised, and proposed methods, we observed an improvement compared to the pre-trained model. Compared to from-scratch training, fine-tuning tends to result in better FID values with the proposed method. This indicates that the proposed method is well-suited for fine-tuning.

Figures 2 and 3 show the samples by fine-tuned models using the unsupervised, supervised, and proposed methods for CIFAR10 (vehicles) and CelebA (male). Since the same random seed was used for both training and generation, the generated images are largely similar. Notably, the images that were sensitive in the unsupervised and supervised methods are replaced by normal images in the proposed method. For example, in Figure 2, images of animals are replaced by images of vehicles, and in Figure 3, images of females are replaced by images of males. While the quality of these images is slightly lower, which may explain why the FID of the proposed method is slightly worse than that of other approaches, the overall image quality of the proposed method remains comparable to other methods. Other results, including Stable Diffusion experiments, are provided in Appendix.

### 5.4.2 Sensitivity of Hyperparameter $\beta$

The proposed method has the hyperparameter $\beta$ defined in Eq. (13), which represents the ratio of sensitive data in the unlabeled data $\mathcal{U}$. Although this can be estimated by using PU learning approaches (Menon et al., 2015; Ramaswamy et al., 2016; Jain et al., 2016; Christoffel et al., 2016; Nakajima & Sugiyama, 2023), increasing $\beta$ amplifies the loss for sensitive data in Eq. (19), which is expected to improve the non-sensitive rate. In this section, we experimentally investigate how adjusting $\beta$ affects the non-sensitive rate and FID.

Figure 4 shows the non-sensitive rate and FID for CIFAR10 animals and vehicles when $\beta$ is set to various values. For CIFAR10 (vehicles), the best non-sensitive rate and FID are achieved when $\beta$ is between $0.15$ and $0.20$. Note that good performance is achieved with a $\beta$ value greater than the true value of $1/11 \approx 0.091$. For CIFAR10 (animals), the best non-sensitive rate is achieved when $\beta = 0.2$, while the best FID is achieved for $\beta \leq 0.091$. These results indicate that adjusting $\beta$ can further improve the performance of the proposed method. Analyzing the optimal value of $\beta$ in the proposed method is our important future work.

## 6 Conclusion

In this paper, we propose positive-unlabeled diffusion models, which prevent diffusion models from generating sensitive data by using unlabeled and sensitive data. With the proposed method, we first incorporate a binary classification framework into diffusion models, and then apply PU learning to this framework. As a result, even without labeled normal data, we can maximize the ELBO for normal data and minimize it for labeled sensitive data, ensuring the generation of only normal data. We performed either from-scratch training or fine-tuning of diffusion models on eight different dataset patterns, and confirmed that the proposed method can improve the non-sensitive rate of generated samples without compromising the FID. We also found that adjusting the hyperparameters can further improve both the non-sensitive rate and FID. In the future, we will focus on analyzing hyperparameters and further improving generation quality.

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
