# OpenReview forum: "Positive-Unlabeled Diffusion Models for Preventing Sensitive Data Generation"
_ICLR.cc/2025/Conference — ICLR 2025 Poster_

### Official Review · Reviewer_o4Dc · 2024-10-23

**Soundness:** 3
**Presentation:** 3
**Contribution:** 2
**Rating:** 6
**Confidence:** 4

**Summary:**

The paper proposes a new scenario in sensitive generation prevention for DMs, where only an unlabeled dataset and a sensitive dataset are available, rather than a fully labeled training dataset. The authors integrate positive-unlabeled learning methods into DMs, and experimental results demonstrate superior performance across multiple datasets.

**Strengths:**

1. The writing is clear and easy to follow.
2. The scenario is new and intriguing in the context of DM's sensitive generation prevention.
3. The experimental results are strong in most cases.

**Weaknesses:**

The weakness lies in the theory part. There is no guarantee of why the proposed method works, making it seem like positive-unlabeled learning methods are directly applied to DMs. Specifically:

In Eq(7), the $L_{DM}$ comes from analyzing the lower bound for $\log p_{\theta}(x_0)$. However, in Eq(13), there is a linear combination of two distributions, and following Eq(7), it is unclear whether this could be decomposed into terms relating to $l_{BCE}$ for the two given distributions (such as finding a lower bound $\log (\beta p_{u}(x) + (1-\beta) p_{N}(x))$). More insights within the theoretical analysis would help validate whether the proposed positive-unlabeled learning methods can indeed be directly implemented as described in the paper.

Besides, there seems to be a few typos, such as it should be \citet in L 253.

**Questions:**

Question:

See weakness. Could the authors provide either proof or assumptions that help explain the proposed methods from the perspective of DM's learning process? I believe the results look promising, and with a solid theoretical analysis, I would consider increasing my score.

---

> ### Author Response · Authors · 2024-11-19
> **Theoretical justification for the proposed method**
>
> Thank you for your feedback.
> We provide a theoretical justification for the proposed method below.
> Since the equations were not rendered properly on OpenReview,
> we have included the same discussion in Appendix A.3 of the Supplementary Material.
>
> ---
>
> We provide a theoretical explanation of the proposed method from the perspective of maximizing and minimizing the ELBO $\mathcal{L}_{\mathrm{DM}}(\mathbf{x};\theta)\approx-\ell(\mathbf{x};\theta)$ defined in Eqs. (7) and (8).
>
> To prevent diffusion models from generating sensitive data,
> we aim to maximize the ELBO for the normal data distribution $p_{\mathcal{N}}(\mathbf{x})$:
>
> $$
> \max_{\theta}(1-\beta)\mathbb{E}_{\mathcal{N}}[-\ell(\mathbf{x};\theta)],
> $$
>
> and minimize it for the sensitive data distribution $p_{\mathcal{S}}(\mathbf{x})$:
>
> $$
> \min_{\theta}\beta\mathbb{E}_{\mathcal{S}}[-\ell(\mathbf{x};\theta)],
> $$
>
> where $\beta$ is the ratio of the sensitive data.
>
> First, we focus on maximizing the ELBO for the normal data.
> As shown in Eqs. (13) and (14),
> we assume that the unlabeled data distribution $p_{\mathcal{U}}(\mathbf{x})$ can be written as a linear combination of $p_{\mathcal{N}}(\mathbf{x})$ and $p_{\mathcal{S}}(\mathbf{x})$:
>
> $$
> p_{\mathcal{U}}(\mathbf{x})=\beta p_{\mathcal{S}}(\mathbf{x})+(1-\beta)p_{\mathcal{N}}(\mathbf{x}) \Leftrightarrow (1-\beta)p_{\mathcal{N}}(\mathbf{x})=p_{\mathcal{U}}(\mathbf{x})-\beta p_{\mathcal{S}}(\mathbf{x}).
> $$
>
> With this assumption, the ELBO for $p_{\mathcal{N}}(\mathbf{x})$ can be rewritten as follows:
>
> $$
> \max_{\theta}(1-\beta)\mathbb{E}_{\mathcal{N}}[-\ell(\mathbf{x};\theta)]=\max_{\theta}\left\{ \mathbb{E}_{\mathcal{U}}[-\ell(\mathbf{x};\theta)]-\beta\mathbb{E}_{\mathcal{S}}[-\ell(\mathbf{x};\theta)]\right\}.
> $$
>
> This maximization equals the following minimization:
>
> $$
> \min_{\theta}(1-\beta)\mathbb{E}_{\mathcal{N}}[\ell(\mathbf{x};\theta)]=\min_{\theta}\left\{ \mathbb{E}_{\mathcal{U}}[\ell(\mathbf{x};\theta)]-\beta\mathbb{E}_{\mathcal{S}}[\ell(\mathbf{x};\theta)]\right\}.
> $$
>
> This is equivalent to the sum of the second and third terms in Eq. (17) of the proposed method,
> where $\ell_{\mathrm{BCE}}(\mathbf{x},0;\theta)=\ell(\mathbf{x};\theta)$.
>
> Next, we focus on minimizing the ELBO for the sensitive data.
> Unfortunately, since $-\ell(\mathbf{x};\theta)$ is unbounded below,
> this minimization diverges to negative infinity,
> leading to meaningless solutions.
> To solve this issue, we introduce the following upper bound for $-\ell(\mathbf{x};\theta)$,
> inspired by the binary classification as shown in Eq. (11):
>
> $$
> -\ell(\mathbf{x};\theta)\leq-\log(1-\exp(-\ell(\mathbf{x};\theta))).
> $$
>
> This upper bound is proportional to $-\ell(\mathbf{x};\theta)$ and approaches zero when $-\ell(\mathbf{x};\theta)$ tends to negative infinity.
> With this upper bound, the ELBO for $p_{\mathcal{S}}(\mathbf{x})$ is bounded above as follows:
>
> $$
> \min_{\theta}\beta\mathbb{E}_{\mathcal{S}}[-\ell(\mathbf{x};\theta)]\leq\min_{\theta}\beta\mathbb{E}_{\mathcal{S}}[-\log(1-\exp(-\ell(\mathbf{x};\theta)))].
> $$
>
> This is equivalent to the first term in Eq. (17),
> where $\ell_{\mathrm{BCE}}(\mathbf{x},1;\theta)=-\log(1-\exp(-\ell(\mathbf{x};\theta)))$.
>
> From the above discussion,
> the proposed method in Eq. (17) can be interpreted as maximizing the ELBO for normal data while minimizing the ELBO for sensitive data.
> For the former,
> we use PU learning to approximate the normal data distribution $p_{\mathcal{N}}(\mathbf{x})$,
> and for the latter, we introduce the upper bound for the ELBO based on the binary classification.
> We will include this discussion in the revised paper.
>
> ---
>
> We will also correct the typos you commented.

---

> > ### Comment · Reviewer_o4Dc · 2024-11-19
> > **Some questions about the ELBO**
> >
> > The proof appears to be mostly correct, except for one part. Please correct me if I have misunderstood.
> >
> > The proof primarily relies on directly expanding the ELBO for normal and sensitive data. However, my concern from the previous review is more focused on the connection between the probability distribution and the ELBO. ELBO should not be used as a direct approximation without caution, particularly when probability distributions undergo linear operations, as it originates from the log-probability. Specifically, the log of a linear combination is not equivalent to the linear combination of log-probabilities.
> >
> > To elaborate:
> >
> > In Equation 7, the log-probability $\log p_{\theta} (x_0) $ is bounded by $ L_{DM}(x_0; \theta) $. However, for the normal data, $p_u $ represents a linear combination, and its log-probability should be expressed as $ \log (\beta p_S (x) + (1 − \beta)p_N (x)) $. I do not see any derivation starting from this term that demonstrates how it extends to the ELBO of $p_S$ and $p_N$. In fact, it seems this issue might not be resolvable, as expanding the log of a linear combination does not straightforwardly lead to a decomposition into individual ELBOs. This log term cannot be omitted, as it directly relates to the diffusion loss.
> >
> > Based on this, I'm quite confused about this linear probability operation. In fact, in guidance-related topics in diffusion models, such as classifier and classifier-free guidance, two probability distributions are often combined using an exponential operation, like $p(x)p(c|x)^{\lambda}$, rather than a linear combination. This is because we are primarily concerned with the log-probability, and only after applying the log operation do we encounter linear computations.
> > .
> >
> > This is my main concern. I look forward to your response.

---

> > > ### Author Response · Authors · 2024-11-19
> > > **Jensen's inequality and our objective**
> > >
> > > Thank you for the quick response. We may not have fully understood the intention of your question, but we will answer from two perspectives.
> > >
> > > First, when $p_{\mathcal{U}}(\mathbf{x})$ can be written as a linear combination of $p_{\mathcal{N}}(\mathbf{x})$ and $p_{\mathcal{S}}(\mathbf{x})$,
> > > the following holds due to Jensen's inequality:
> > >
> > > $$
> > > \log p_{\mathcal{U}}(\mathbf{x})=\log\left(\beta p_{\mathcal{S}}(\mathbf{x})+(1-\beta)p_{\mathcal{N}}(\mathbf{x})\right)\geq\beta\log p_{\mathcal{S}}(\mathbf{x})+(1-\beta)\log p_{\mathcal{N}}(\mathbf{x}).
> > > $$
> > >
> > > As you commented, $\log p_{\mathcal{U}}(\mathbf{x})$ is not equal to a linear combination of $\log p_{\mathcal{N}}(\mathbf{x})$ and $\log p_{\mathcal{S}}(\mathbf{x})$, but they are related by this inequality.
> > > If we were to consider the ELBOs for these distributions, we could use this relationship.
> > >
> > > Next, this paper focuses on the ELBO for the diffusion model $p_{\theta}(\mathbf{x})$  and does not consider the ELBO for the true distributions $p_{\mathcal{N}}(\mathbf{x})$, $p_{\mathcal{S}}(\mathbf{x})$, and $p_{\mathcal{U}}(\mathbf{x})$.
> > >
> > > Our goal is to make the diffusion model $p_{\theta}(\mathbf{x})$ similar to $p_{\mathcal{N}}(\mathbf{x})$ while keeping it far from $p_{\mathcal{S}}(\mathbf{x})$.
> > > This is the same as optimizing the following two KL divergences:
> > >
> > > $$
> > > \min_{\theta}D_{KL}(p_{\mathcal{N}}||p_{\theta})\approx\max_{\theta}\mathbb{E}_{\mathcal{N}}[-\ell(\mathbf{x};\theta)],
> > > $$
> > >
> > > and
> > >
> > > $$
> > > \max_{\theta}D_{KL}(p_{\mathcal{S}}||p_{\theta})\approx\min_{\theta}\mathbb{E}_{\mathcal{S}}[-\ell(\mathbf{x};\theta)].
> > > $$
> > >
> > > The important point is that in the optimization above, we are simply calculating the expectation of the ELBO $-\ell(\mathbf{x};\theta)$ using the true distributions $p_{\mathcal{N}}(\mathbf{x})$ and $p_{\mathcal{S}}(\mathbf{x})$.
> > > In other words, there is no need to consider the logarithms of the true distributions, $\log p_{\mathcal{N}}(\mathbf{x})$ and $\log p_{\mathcal{S}}(\mathbf{x})$.
> > >
> > >
> > > Does the explanation above address your question? If there is anything unclear or of concern, please feel free to let us know.

---

> > > > ### Comment · Reviewer_o4Dc · 2024-11-19
> > > > **Concerns Addressed**
> > > >
> > > > Thanks for the quick reply! I hope this part could also be shown in the final ver. I raise my score to 6.

---

> > > > > ### Author Response · Authors · 2024-11-19
> > > > >
> > > > > Thank you for the quick reply. I will organize this discussion and include it in the final version later. Thank you very much!

---

### Official Review · Reviewer_mThp · 2024-10-31

**Soundness:** 2
**Presentation:** 2
**Contribution:** 2
**Rating:** 5
**Confidence:** 5

**Summary:**

This paper proposes a novel method to prevent the generation of sensitive images from Diffusion Models. It can estimate the evidence lower bound for normal images without using labeled normal images. Experiments show that the proposed method can alleviate the generation of sensitive images while maintaining high image quality.

**Strengths:**

This paper has the following strengths:
- It is helpful for estimating the evidence lower bound for normal images without relying on labeled examples.
- The writing of the method part is clear which makes the method easy to understand.

**Weaknesses:**

This paper has the following weaknesses:
- The presentation of qualitative examples is not very good. The images in Figure 2 and Figure 3 are small and blurry. Besides, as mentioned by the authors, "Since the same random seed was used for both training and generation, the generated images are largely similar". It is harder to see the difference clearly.
- This paper only tests the performance of the proposed method. It does not have a comparison of previous closely related works.
- The setting/scenario is not very close to real-world **sensitive** images. For example, it treats even numbers as normal while odd numbers as sensitive. It would be better to consider more real-world common scenarios/settings as in previous works [1, 2].
- It does not conduct experiments using popular SD such as SD v 1.4, SD v2.0, and SD XL.

--------
**References**.
[1] Selective Amnesia: A Continual Learning Approach to Forgetting in Deep Generative Models. In *NeurIPS 2023*.
[2] Erasing Concepts from Diffusion Models. In *ICCV 2023*.

**Questions:**

The questions/suggestions are mainly based on weaknesses.
- Could authors compare the performance of the proposed method with the works in the related work section?
- Could authors conduct experiments using SD v1.4 and SD v2.0?
- It would be better to contain more clear images in the Appendix.
- Some sentences in the abstract are not very clear. For example, 'Our approach can approximate the evidence lower bound
(ELBO) for normal (negative) data using only unlabeled and sensitive (positive) data.' Does this mean the sensitive data are unlabeled? But the abstract also says 'we address this problem by using a small amount of labeled sensitive data.'

---

> ### Author Response · Authors · 2024-11-25
> **Rebuttal**
>
> Thank you for your feedback.
> We apologize for the delayed response;
> the experiments with Stable Diffusion took longer than expected.
>
>
> > Weakness: This paper only tests the performance of the proposed method. It does not have a comparison of previous closely related works.
>
> > Question: Could authors compare the performance of the proposed method with the works in the related work section?
>
> We did not compare our approach with existing methods such as selective amnesia (SA) (Heng \& Soh, 2024) and erasing stable diffusion (ESD) (Gandikota et al., 2023) for the following two reasons.
>
> The first reason is that existing methods are not designed for the Positive-Unlabeled (PU) setting.
> For example, in the case of SA, it assumes that all unlabeled data are normal in the PU setting, maximizing the ELBO in Eq. (7) for the unlabeled data and minimizing the ELBO for the sensitive data.
> However, since the unlabeled data actually contain sensitive data, this approach inadvertently maximizes the ELBO for the sensitive data, encouraging the generation of sensitive data.
> The same applies to the case of ESD.
>
> The second reason is that the supervised diffusion models (SDM) proposed in section 3.1 cover the functionality of the existing methods.
> Although SA aims to maximize the ELBO for the unlabeled data and minimize it for the sensitive data, it instead optimizes the surrogate objective function as explained in Section 4.1 to avoid instability issues during training.
> On the other hand, SDM achieves stable training with the original objective of SA, which maximizes the ELBO for unlabeled data and minimizes it for sensitive data.
> Therefore, SDM can be seen as a generalization of SA.
> In the experimental section, we demonstrate that our approach outperforms SDM in the PU setting.
>
>
> > Weakness: The setting/scenario is not very close to real-world sensitive images ...
>
> > Weakness: It does not conduct experiments using popular SD such as SD v 1.4, SD v2.0, and SD XL.
>
> > Questions: Could authors conduct experiments using SD v1.4 and SD v2.0?
>
> We conducted experiments using Stable Diffusion 1.4 with the following experimental settings.
> The objective was to ensure that specific individuals (in this case, Brad Pitt) are excluded when generating images of middle-aged men using Stable Diffusion.
> The dataset was prepared as follows:
>
> - Unlabeled data: 64 images of middle-aged men and 16 images of Brad Pitt.
> - Labeled sensitive data: 20 images of Brad Pitt.
>
> These images were generated using Stable Diffusion with the prompts "a photo of a middle-aged man" and "a photo of Brad Pitt".
> This experimental setup is an extension of (Heng \& Soh, 2024) to the PU setting.
>
> Using this dataset,
> we applied standard fine-tuning (unsupervised),
> supervised diffusion models as described in Section 3.1 (supervised),
> and the proposed method to Stable Diffusion 1.4.
>
> As an evaluation metric,
> we use the non-sensitive rate described in Section 5.2,
> which represents the ratio of the generated samples classified as normal (non-sensitive) by a pre-trained classifier.
> For the pre-trained classifier,
> we use a ResNet-34 fine-tuned on 1,000 images each of middle-aged men and Brad Pitt, generated by Stable Diffusion.
> The accuracy on the test set for the classification task is 99.75\%.
>
> The experimental setup is almost the same as in Section 5.3,
> with a batch size of 16, a learning rate of $10^{-5}$ for the proposed method and $10^{-4}$ for the others,
> 1,000 epochs,
> and $\beta=0.2$ for the proposed method.
>
> The experimental results for generating images of middle-aged men are provided in Appendix A.6.
> With the Unsupervised and Supervised methods,
> attempts to generate images of middle-aged men often result in the generation of Brad Pitt.
> This issue arises due to the presence of Brad Pitt in the unlabeled data.
> Meanwhile, the proposed method successfully suppresses the generation of Brad Pitt.
>
> The non-sensitive rates are as follows:
> | Unsupervised | Supervised | Proposed |
> |--------------|------------|----------|
> | 0.80         | 0.84       | 0.99     |
>
> Quantitatively,
> the proposed method demonstrates superior performance compared to both the Unsupervised and Supervised methods.
>
>
> > Weakness: The presentation of qualitative examples is not very good ...
>
> > Question: It would be better to contain more clear images in the Appendix.
>
> We have included higher-resolution experimental results using Stable Diffusion in Appendix A.6.
> We also plan to include higher-resolution results for other experiments as well.
>
>
> > Question: Some sentences in the abstract are not very clear ...
>
> We apologize for the confusion.
> The sentence you pointed out should be more accurately stated as follows:
>
> "Our approach can approximate the evidence lower bound (ELBO) for normal (negative) data using only unlabeled data and labeled sensitive (positive) data."
>
> We will revise the abstract accordingly.

---

> > ### Comment · Reviewer_mThp · 2024-11-25
> > **Response to Authors**
> >
> > Thanks for your response. My major concern has been addressed. I have raised my score. I hope the authors will also add those to the final version.

---

> ### Author Response · Authors · 2024-11-25
>
> Thank you for the quick reply. I will include this discussion in the final version. Thank you very much!

---

### Official Review · Reviewer_KfYa · 2024-11-03

**Soundness:** 2
**Presentation:** 3
**Contribution:** 3
**Rating:** 5
**Confidence:** 3

**Summary:**

This paper presents positive-unlabeled diffusion models to address the challenge of generating sensitive data by diffusion models. Traditional training methods often lead to the inclusion of unwanted sensitive data, prompting the authors to propose a novel approach that utilizes a small amount of labeled sensitive data alongside unlabeled data, which may contain both normal and sensitive instances. The core idea is to maximize the evidence lower bound (ELBO) for normal data using only unlabeled and sensitive data, enabling the generation of only normal content. By treating the unlabeled data distribution as a mixture of normal and sensitive distributions, the approach allows effective model training without needing extensive labeled normal data.

The overall logic of the paper is coherent, providing a detailed introduction to the relevant technologies and concepts. In particular, the work appears to be well-motivated and the methodology is reasonable. However, the rationale for the proposed method and the challenges associated with it, as well as comparisons with other methods, are not thoroughly explained. For instance, the paper does not clarify why prior methods cannot be directly adapted to diffusion models. The underlying reasons for the unique characteristics of the diffusion model or the challenges inherent in the adaptation process remain unaddressed.

**Strengths:**

- Paper studies an important topic that is highly related to diffusion model and privacy protection.
- Paper is easy to follow.

**Weaknesses:**

- The methodological details, and the rationale for adopting the methods, are rather confusing.
- limited contribution and novelty.
- The selection of datasets for the experiments and the design of the experimental methods seem to be unreasonable.

**Questions:**

1. In Section 4, the disadvantages of previous work and the advantages of the proposed method seem to be inadequately explained and presented. For instance, in Section 4.1, the statement "Moreover, it cannot work well in the PU setting because it assumes all unlabeled data are normal" raises the question of what specific issues arise from assuming that all unlabeled data are normal, yet this point is not well articulated.
2. In Section 4.2, a work related to "semi-supervised anomaly detection" is mentioned, which appears to be highly relevant to the proposed method. However, the paper does not clarify why this work cannot be directly adapted to diffusion models. What are the challenges and difficulties associated with this transition?
3. The experiments involve selecting several classic datasets and categorizing them into two classes. However, the rationale for these datasets being sufficiently representative of sensitive and non-sensitive data is unclear. Although these are binary classification tasks, the boundaries between sensitive and non-sensitive data do not seem to be so distinct. Additionally, in Section 5.4.1 on page 9, it states, "since the ratio of normal data to sensitive data in the unlabeled data U is 10:1." Please clarify the reasoning behind this choice of ratio.

**Details Of Ethics Concerns:**

None.

---

> ### Author Response · Authors · 2024-11-21
> **Rebuttal (1/2)**
>
> Thank you for your feedback, which we shall address below.
>
> > Weakness: The methodological details, and the rationale for adopting the methods, are rather confusing.
>
> We apologize for the confusion.
> In this paper, we derive the proposed method from the perspective of PU learning:
> first, we introduce the ELBO for the diffusion model as described in Eq. (8);
> next, we define a binary classifier using this ELBO as described in Eq. (10);
> and finally, we apply PU learning to this binary classifier as described in Eq. (17).
> Through the training of this binary classifier,
> the ELBO is maximized for normal data and minimized for sensitive data.
>
> In addition,
> we can provide the theoretical justification for the proposed method from the perspective of maximizing and minimizing the ELBO,
> as described in the rebuttal for Reviewer o4Dc.
> Note that we have included the same discussion in Appendix A.3 of the Supplementary Material.
>
>
> > Weakness: limited contribution and novelty. & Question 1
>
> We first note that we have included the following discussion in Appendix A.5 of the Supplementary Material since the equations were not rendered properly on OpenReview.
>
> We apologize for the insufficient explanation.
> Existing methods assume that all unlabeled data are normal,
> and maximize the ELBO $\mathcal{L}_{\mathrm{DM}}(\mathbf{x};\theta)$ in Eq. (7) for the unlabeled data.
>
> Let $p_{\mathcal{U}}(\mathbf{x})$, $p_{\mathcal{N}}(\mathbf{x})$, $p_{\mathcal{S}}(\mathbf{x})$ represent the unlabeled, normal and sensitive data distribution, respectively.
>
> By using $p_{\mathcal{N}}(\mathbf{x})$ and $p_{\mathcal{S}}(\mathbf{x})$,
> $p_{\mathcal{U}}(\mathbf{x})$ can be rewritten as follows:
> \begin{align}
>   p_{\mathcal{U}}(\mathbf{x})&=\beta p_{\mathcal{S}}(\mathbf{x})+(1-\beta)p_{\mathcal{N}}(\mathbf{x}),
> \end{align}
> where $\beta$ is the ratio of the sensitive data.
>
> Hence, the ELBO for the unlabeled data can be rewritten as:
>
> \begin{align}
>   \max_{\theta}\mathbb{E}_{p_{\mathcal{U}}}[\mathcal{L}_{\mathrm{DM}}(\mathbf{x};\theta)]=\max_{\theta}\left\{ \beta\mathbb{E}_{p_{\mathcal{S}}}[\mathcal{L}_{\mathrm{DM}}(\mathbf{x};\theta)]+(1-\beta)\mathbb{E}_{p_{\mathcal{N}}}[\mathcal{L}_{\mathrm{DM}}(\mathbf{x};\theta)]\right\}.
> \end{align}
>
> Therefore, if all unlabeled data are assumed to be normal,
> the ELBO will be maximized even for the sensitive data included in the unlabeled data.
> This would inadvertently encourage the generation of sensitive data.
>
> Meanwhile, our approach maximizes the ELBO for the normal data by using the unlabeled and sensitive data as follows:
>
> \begin{align}
> \max_{\theta}(1-\beta)\mathbb{E}_{p_{\mathcal{N}}}[\mathcal{L}_{\mathrm{DM}}(\mathbf{x};\theta)]=\max_{\theta}\left\{ \mathbb{E}_{p_{\mathcal{U}}}[\mathcal{L}_{\mathrm{DM}}(\mathbf{x};\theta)]-\beta\mathbb{E}_{p_{\mathcal{S}}}[\mathcal{L}_{\mathrm{DM}}(\mathbf{x};\theta)]\right\}.
> \end{align}
>
> Therefore, our approach can maximize the ELBO for normal data without using labeled normal data.
>
>
> > Question 2
>
> We apologize for the insufficient explanation.
> Although semi-supervised anomaly detection is highly relevant to our approach,
> it cannot be directly applied to diffusion models.
> The reasons are explained below.
>
> Semi-supervised anomaly detection trains an anomaly detector with parameter $\phi$ by minimizing the anomaly score $\ell_{\mathrm{AD}}(\mathbf{x}; \phi)$ for normal data within unlabeled data,
> and maximizing it for the labeled anomaly data.
> $\ell_{\mathrm{AD}}(\mathbf{x}; \phi)$ satisfies $0 \leq \ell_{\mathrm{AD}}(\mathbf{x}; \phi) < \infty$.
> For example, [1] minimizes $\ell_{\mathrm{AD}}(\mathbf{x}; \phi)$ for unlabeled data,
> and minimizes $1 / \ell_{\mathrm{AD}}(\mathbf{x}; \phi)$ for labeled anomaly data.
>
> In contrast, to prevent sensitive data generation in diffusion models,
> it is necessary to maximize the log-likelihood for normal data,
> and minimize it for sensitive data.
> However, since the log-likelihood is unbounded below, minimizing it leads to divergence to negative infinity, resulting in meaningless solutions.
> Therefore, existing semi-supervised anomaly detection approaches like [1] cannot be directly applied to diffusion models.
>
> To solve this problem,
> in Section 3.1,
> we define a binary classifier $p_{\theta}(y|\mathbf{x})$ by using the ELBO of the log-likelihood defined in Eq. (8),
> according to ABC (Yamanaka et al., 2019).
> $-\log p_{\theta}(y|\mathbf{x})$ satisfies $0 \leq -\log p_{\theta}(y|\mathbf{x}) < \infty$,
> enabling stable maximization and minimization of the log-likelihood through the training of this binary classifier.
>
> In summary, our challenge lies in the definition of $-\log p_{\theta}(y|\mathbf{x})$,
> which corresponds to the anomaly score $\ell_{\mathrm{AD}}(\mathbf{x}; \phi)$ in semi-supervised anomaly detection.
> By using this loss function,
> diffusion models can be incorporated into the existing semi-supervised anomaly detection framework.
>
> [1] Ruff, Lukas, et al. "Deep semi-supervised anomaly detection." ICLR2020.

---

> ### Author Response · Authors · 2024-11-21
> **Rebuttal (2/2)**
>
> > Weakness: The selection of datasets for the experiments and the design of the experimental methods seem to be unreasonable. & Question 3
>
> In practical applications,
> sensitive content may be nude images of a particular individual.
> In such cases, the sensitive data correspond to facial images of that individual,
> while the non-sensitive data correspond to facial images of other individuals.
> Since the sensitive data are less diverse and the boundary between sensitive and non-sensitive data is clear in this scenario,
> it is relatively easy to prevent the generation of sensitive data.
>
> On the other hand, in our experiments,
> we assume scenarios where sensitive data are more diverse.
> For example, in the CelebA experiment,
> we assume that male faces are normal while female faces are sensitive.
> Female faces are highly diverse, and distinguishing them from male faces can sometimes be challenging.
> The goal of our experiments is to evaluate how effective our approach is in these more challenging scenarios.
>
> Additionally,
> since there has been no existing research on training diffusion models in the PU setting,
> we set the ratio of normal data to sensitive data in our experiments based on the semi-supervised anomaly detection experiments (Takahashi et al., 2024),
> which is discussed as a related work in Section 4.2.
>
> About comparisons with other methods,
> we did not compare our approach with existing methods such as selective amnesia (SA) (Heng \& Soh, 2024) for the following two reasons.
>
> The first reason is that existing methods are not designed for the Positive-Unlabeled (PU) setting,
> as described in rebuttal for Question 1.
>
> The second reason is that the supervised diffusion models (SDM) proposed in section 3.1 cover the functionality of the existing methods.
> Although SA aims to maximize the ELBO for the unlabeled data and minimize it for the sensitive data, it instead optimizes the surrogate objective function as explained in Section 4.1 to avoid instability issues during training.
> On the other hand, SDM achieves stable training with the original objective of SA, which maximizes the ELBO for unlabeled data and minimizes it for sensitive data.
> Therefore, SDM can be seen as a generalization of SA.
> In the experimental section, we demonstrate that our approach outperforms SDM in the PU setting.

---

> ### Author Response · Authors · 2024-11-25
> **Additional Experiments with Stable Diffusion 1.4**
>
> To address the third Weakness, Question 3, and feedback from other reviewer (mThp),
> we conducted following experiments using Stable Diffusion 1.4.
>
> The objective was to ensure that specific individuals (in this case, Brad Pitt) are excluded when generating images of middle-aged men using Stable Diffusion.
> The dataset was prepared as follows:
>
> - Unlabeled data: 64 images of middle-aged men and 16 images of Brad Pitt.
> - Labeled sensitive data: 20 images of Brad Pitt.
>
> These images were generated using Stable Diffusion with the prompts "a photo of a middle-aged man" and "a photo of Brad Pitt".
> This experimental setup is an extension of (Heng \& Soh, 2024) to the PU setting.
>
> Using this dataset,
> we applied standard fine-tuning (unsupervised),
> supervised diffusion models as described in Section 3.1 (supervised),
> and the proposed method to Stable Diffusion 1.4.
>
> As an evaluation metric,
> we use the non-sensitive rate described in Section 5.2,
> which represents the ratio of the generated samples classified as normal (non-sensitive) by a pre-trained classifier.
> For the pre-trained classifier,
> we use a ResNet-34 fine-tuned on 1,000 images each of middle-aged men and Brad Pitt, generated by Stable Diffusion.
> The accuracy on the test set for the classification task is 99.75\%.
>
> The experimental setup is almost the same as in Section 5.3,
> with a batch size of 16, a learning rate of $10^{-5}$ for the proposed method and $10^{-4}$ for the others,
> 1,000 epochs,
> and $\beta=0.2$ for the proposed method.
>
> The experimental results for generating images of middle-aged men are provided in Appendix A.6.
> With the Unsupervised and Supervised methods,
> attempts to generate images of middle-aged men often result in the generation of Brad Pitt.
> This issue arises due to the presence of Brad Pitt in the unlabeled data.
> Meanwhile, the proposed method successfully suppresses the generation of Brad Pitt.
>
> The non-sensitive rates are as follows:
> | Unsupervised | Supervised | Proposed |
> |--------------|------------|----------|
> | 0.80         | 0.84       | 0.99     |
>
> Quantitatively,
> the proposed method demonstrates superior performance compared to both the Unsupervised and Supervised methods.

---

### Official Review · Reviewer_JZxJ · 2024-11-04

**Soundness:** 2
**Presentation:** 3
**Contribution:** 3
**Rating:** 8
**Confidence:** 3

**Summary:**

This work proposes positive-unlabeled diffusion models that prevent the generation of sensitive data using unlabeled and sensitive data. To address the issue that labeling all sensitive data in the large-scale unlabeled training data is impractical, the authors propose to use a small amount of labeled sensitive data and then propose an approach based on PU learning to train a binary classifier to distinguish between positive and negative data. Simulation results show that the proposed method does effectively prevent generation of sensitive data. Interesting work.

**Strengths:**

See above summary.

**Weaknesses:**

I wouldn't count it as a weakness, but more like a further discussion.

**Questions:**

I assume that for this method to work, the small amount of labeled sensitive data should well represent the characteristics/properties of all sensitive data. I am just wondering what will happen if there is a mismatch? For example, in the MNIST example that the authors show in the paper, what if the labeled sensitive dataset includes some but not all odd numbers?

---

> ### Author Response · Authors · 2024-11-19
> **Limitations of our approach**
>
> Thank you for your feedback.
> As you commented, our approach requires that the labeled sensitive data represent the characteristics of all sensitive data.
> We will explain this using the example of MNIST,
> where even numbers are considered normal and odd numbers are considered sensitive.
> If the unlabeled data contain all even and odd numbers,
> but the labeled sensitive data only include 1, 3, 5, and 7,
> then the proposed method would generate the digit 9.
>
> To solve this issue, it is necessary to prepare a sufficient variety of labeled sensitive data.
> In the MNIST example, it would be enough to include the digit 9 in the labeled sensitive data.
> Similarly, if we want to prevent the generation of a face of a particular person,
> it would be enough to prepare just a few photos of that person.
> However, if the sensitive data are more diverse, such as male or female faces, or categories like vehicles or animals,
> then our approach would likely require a larger amount of labeled sensitive data.
> In the experiments using STL10 and CelebA,
> we successfully prevented the generation of sensitive data with 200 labeled sensitive samples.
> Since the faces, vehicles, and animals included in the training and test sets are different,
> we believe our approach demonstrated a certain level of generalization ability.
>
> In any case, addressing this issue is an important part of our future work.
> We plan to explore ways to prevent the generation of sensitive data not covered by the labeled sensitive data,
> for example,
> by using supplementary information such as text descriptions.
> We have included these limitations in Appendix A.4 of the Supplementary Material.

---

### Comment · Area_Chair_ZVAa · 2024-11-24

Dear reviewers,

Thanks for serving as a reviewer. As the discussion period comes to a close and the authors have submitted their rebuttals, I kindly ask you to take a moment to review them and provide any final comments.

If you have already updated your comments, please disregard this message.

Thank you once again for your dedication to the OpenReview process.

Best,

Area Chair

---

### Meta-Review · Area_Chair_ZVAa · 2024-12-21

**Metareview:**

This work proposes positive-unlabeled diffusion models that prevent the generation of sensitive data using unlabeled and sensitive data. To address the issue that labeling all sensitive data in the large-scale unlabeled training data is impractical, the authors propose to use a small amount of labeled sensitive data and then propose an approach based on PU learning to train a binary classifier to distinguish between positive and negative data. Simulation results show that the proposed method does effectively prevent generation of sensitive data. Interesting work.

Strength:

1. The proposed scenario is important and intriguing in the context of DM's sensitive generation prevention.

2. Empirical results demonstrate it effectiveness.

Weaknesses:

Still lacking full evaluations on current state-of-the-art Text-to-Image models like SDXL. This somehow limit the paper's impact.

In summary, I think most concerns have been addressed during the rebuttal phase and this paper can be accepted.

**Additional Comments On Reviewer Discussion:**

The authors and reviewers discuss about the evaluation setting, theoretical analysis, etc. These discussions greatly improve the paper. Most reviewers say their concerns have been addressed except one does not response. However, after reading the authors response, I think the reviewers' concerns have already be addressed.

---

### Decision · Program_Chairs · 2025-01-22

Accept (Poster)